# The Potential of Ultrasound Radiomics in Carpal Tunnel Syndrome Diagnosis: A Systematic Review and Meta-Analysis

**DOI:** 10.3390/diagnostics13203280

**Published:** 2023-10-23

**Authors:** Wei-Ting Wu, Che-Yu Lin, Yi-Chung Shu, Peng-Chieh Shen, Ting-Yu Lin, Ke-Vin Chang, Levent Özçakar

**Affiliations:** 1Department of Physical Medicine and Rehabilitation, National Taiwan University Hospital, College of Medicine, National Taiwan University, Taipei 10048, Taiwan; wwtaustin@yahoo.com.tw; 2Department of Physical Medicine and Rehabilitation, National Taiwan University Hospital, Bei-Hu Branch, Taipei 10845, Taiwan; 3Institute of Applied Mechanics, College of Engineering, National Taiwan University, Taipei 10617, Taiwan; cheyu@ntu.edu.tw (C.-Y.L.); yichung@iam.ntu.edu.tw (Y.-C.S.); 4Department of Physical Medicine and Rehabilitation, Lo-Hsu Medical Foundation, Inc., Lotung Poh-Ai Hospital, Yilan 26546, Taiwan; jamesshen98360315@gmail.com (P.-C.S.); t840326@yahoo.com.tw (T.-Y.L.); 5Center for Regional Anesthesia and Pain Medicine, Wang-Fang Hospital, Taipei Medical University, Taipei 11600, Taiwan; 6Department of Physical and Rehabilitation Medicine, Hacettepe University Medical School, Ankara 06100, Turkey; lozcakar@yahoo.com

**Keywords:** median nerve, wrist, ultrasonography, artificial intelligence, machine learning

## Abstract

**Background**: Carpal tunnel syndrome (CTS) is the most common entrapment neuropathy for which ultrasound imaging has recently emerged as a valuable diagnostic tool. This meta-analysis aims to investigate the role of ultrasound radiomics in the diagnosis of CTS and compare it with other diagnostic approaches. **Methods**: We conducted a comprehensive search of electronic databases from inception to September 2023. The included studies were assessed for quality using the Quality Assessment Tool for Diagnostic Accuracy Studies. The primary outcome was the diagnostic performance of ultrasound radiomics compared to radiologist evaluation for diagnosing CTS. **Results**: Our meta-analysis included five observational studies comprising 840 participants. In the context of radiologist evaluation, the combined statistics for sensitivity, specificity, and diagnostic odds ratio were 0.78 (95% confidence interval (CI), 0.71 to 0.83), 0.72 (95% CI, 0.59 to 0.81), and 9 (95% CI, 5 to 15), respectively. In contrast, the ultrasound radiomics training mode yielded a combined sensitivity of 0.88 (95% CI, 0.85 to 0.91), a specificity of 0.88 (95% CI, 0.84 to 0.92), and a diagnostic odds ratio of 58 (95% CI, 38 to 87). Similarly, the ultrasound radiomics testing mode demonstrated an aggregated sensitivity of 0.85 (95% CI, 0.78 to 0.89), a specificity of 0.80 (95% CI, 0.73 to 0.85), and a diagnostic odds ratio of 22 (95% CI, 12 to 41). **Conclusions**: In contrast to assessments by radiologists, ultrasound radiomics exhibited superior diagnostic performance in detecting CTS. Furthermore, there was minimal variability in the diagnostic accuracy between the training and testing sets of ultrasound radiomics, highlighting its potential as a robust diagnostic tool in CTS.

## 1. Introduction

Ultrasound radiomics stands as a cutting-edge medical imaging methodology, focusing on the meticulous extraction and analyses of an extensive array of quantitative features sourced from ultrasound images [1]. These features are systematically acquired from the images, aiming to encapsulate intricate nuances, patterns, and textures that may elude the capabilities of human visual perception. For instance, its efficacy for predicting microvascular invasion in hepatocellular carcinoma [2], forecasting the lymphovascular invasion in patients with invasive breast cancer [3], and envisaging the likelihood of extensive cervical lymph node metastasis in papillary thyroid carcinoma have been demonstrated [4]. Herewith, ultrasound radiomics presents some challenges as well. These include the imperative need for standardized image acquisition protocols, the discerning selection of pertinent features, and the establishment of robust analytical pipelines. The domain of radiomics holds the potential to confer a multitude of advantages upon image processing in the realm of medical imaging [5]. These advantages encompass the mitigation of subjectivity inherent in manual image interpretation and the embrace of a data-driven approach. Such an approach possesses the capability to unveil latent patterns, correlations, and associations concealed within imaging data.

Carpal tunnel syndrome (CTS) stands as the most prevalent entrapment neuropathy, with reported prevalence rate of 14.4% in the general population [6]. It is paramount not to underestimate the disease burden in CTS, particularly among individuals of working age. Previous surveys have illuminated the adverse outcomes experienced by claimants, including job changes and work cessation coupled with time loss due to CTS [7]. While non-surgical and surgical approaches alike have evolved in the treatment of CTS, the crux of the matter lies in the diagnostic accuracy. Electrophysiological investigations conventionally serve as the gold standard for diagnosing CTS. Patient discomfort during these examinations should not be disregarded though. 

In recent years, the utility of ultrasound in assessing entrapment neuropathies has garnered recognition [8], and its diagnostic accuracy in comparison to electrophysiological studies has been extensively explored. A spectrum of ultrasound modalities has been employed in diagnosing CTS, ranging from traditional grayscale imaging [9] to cutting-edge sonoelastography [10]. Notably, there is a growing body of research pertaining to ultrasound radiomics for CTS diagnosis. The potential benefit of ultrasound radiomics in diagnosing CTS lies in its capacity to extract and analyze intricate image features, thereby providing a more detailed and objective assessment of the condition when compared to the sole estimation of the nerve’s cross-section or shape. Due to the lack of previous meta-analyses on this subject, our objective was to investigate the diagnostic accuracy of ultrasound radiomics in detecting CTS and to compare it with other ultrasound diagnostic approaches. The meta-analysis would aid in validating whether radiomics could improve the diagnostic efficacy of standard ultrasound imaging for assessing CTS, thereby making a significant contribution to the field of neuromuscular pain medicine.

## 2. Methods

### 2.1. Protocol Registration 

This meta-analysis adhered to the PRISMA 2020 guidelines (Appendix A) and was preceded by the registration of the study protocol on inplasy.com accessed on 20 September 2023 under registration number (INPLASY202390069).

### 2.2. Search Strategy 

To locate potentially eligible research articles, we conducted a systematic search across three electronic databases, namely PubMed, Embase, and Web of Science (from their inception to September 2023). The search criteria included the following keywords: (“radiomics” OR “radiomic” OR “ultrasound radiomics”) AND (“carpal tunnel syndrome” OR “median nerve” OR “neuropathy” OR “nerve”). We also reviewed the reference lists of the located articles by hand to discover any additional relevant studies. Importantly, this search was carried out without any language restriction. Furthermore, we explored ClinicalTrials.gov to retrieve unpublished data from ongoing trials. For a comprehensive description of the search strategy (Appendix A).

### 2.3. Inclusion and Exclusion Criteria 

The PICO (population, intervention, comparison, and outcome) question was structured in the following manner: (1) Population: Individuals diagnosed with CTS; (2) Intervention: Application of ultrasound radiomics; (3) Comparison: Evaluation of ultrasound imaging for the diagnosis of CTS by radiologists; and (4) Outcome: Assessment of diagnostic performance indicators, such as sensitivity and specificity. In the context, ultrasound radiomics is a quantitative and data-driven approach that involves extracting and analyzing a wide range of features from ultrasound images to provide valuable insights into tissue characteristics and potential disease markers. Furthermore, in our mete-analysis, we did not specify any specific types of ultrasound radiomics or radiologist’s evaluation.

Our selection criteria encompassed clinical studies centered on the utilization of ultrasound radiomics for the diagnosis of CTS. We excluded articles that employed ultrasound radiomics not for diagnostic purposes or those that exclusively delved into deep learning techniques without referencing radiomics. Additionally, our exclusion criteria comprised case reports, letters, editorials, commentaries, posters, and unpublished articles.

To ensure the rigor of our review process, two independent authors, W.-T.W. and T.-Y.L., initially conducted an evaluation of the titles and abstracts of the papers. Eligible articles underwent a comprehensive examination in their entirety to confirm their alignment with all the stipulated inclusion and exclusion criteria. The data extraction was performed independently by two authors. In the initial stage of data extraction, strict consistency was not mandated. However, in cases where inconsistencies arose regarding specific items, these were resolved through discussion between the two authors. Additionally, any unresolved discrepancies were reported to the corresponding author for a final decision. In the event of missing or incomplete data, our approach would involve reaching out to the corresponding author of the referenced paper. Should we not receive a response from the corresponding authors, we would document this situation in Appendix A.

### 2.4. Quality Assessment

Two independent authors assessed the study quality using the Quality Assessment of Diagnostic Accuracy Studies (QUADAS)-2 tool [11], which contains four critical domains: (1) patient selection, (2) index test, (3) reference standard, and (4) flow and timing of the primary studies integrated into the meta-analysis. In each of these domains, the potential for bias was classified as high, low, or unclear. Discrepancies in the quality assessment were resolved through a consensus-based approach, mirroring the procedure followed for data extraction. It is noteworthy that the outcome of quality assessment did not influence the decision to include or exclude a particular study in this meta-analysis. Instead, the determination regarding the inclusion of an article was made based on the pre-established inclusion and exclusion criteria.

### 2.5. Statistical Analysis 

In conducting our statistical analyses, Stata software (StataCorp 2015, Stata Statistical Software: Release 14, StataCorp LP, College Station, TX, USA) was employed. Statistical significance was assumed when the *p*-value fell below the threshold of 0.05. As the primary objective of this study was to evaluate the efficacy of ultrasound radiomics or radiologist’s evaluation in diagnosing CTS, sensitivity, specificity, and diagnostic odds ratio were analyzed using a random-effects model. Sensitivity assesses the diagnostic test’s capacity to accurately detect individuals, such as those assessed using ultrasound radiomics or radiologist’s evaluation, who have a specific condition or disease, like the presence of CTS. It quantifies the proportion of true positive results among all individuals who genuinely have the condition. In contrast, specificity gauges the test’s ability to correctly identify individuals without the condition, indicating the proportion of true negative results among all individuals who are indeed free of the condition. 

The data chosen for consolidation in each study comprised results demonstrating the highest diagnostic performance within their particular category. For instance, when assessing diagnostic performance (evaluated via the area under the Receiver Operating Characteristic curve analysis) in the context of radiologist evaluations, if Radiologist A outperformed Radiologist B, the dataset from Radiologist A was selected for further analysis. To gauge the diagnostic performance, summary receiver operating characteristics (sROC) analysis was employed to determine the area under the curve (AUC). Additionally, heterogeneity was assessed using I^2^ statistics, considering significant heterogeneity when I^2^ exceeded 50% [12]. To address the potential for publication bias, the Deeks’ funnel plot asymmetry test was conducted [13].

## 3. Results

### 3.1. Literature Search 

The search identified 109 publications. After reading their title/abstract, 62 articles including the duplicates were discarded. Full texts of the remaining 10 studies were then assessed carefully. An additional five articles were excluded for the following reasons: one focused on using magnetic resonance imaging for characterizing CTS [14], another analyzed the optic nerve [15], one targeted nerve block [16], one emphasized differentiating demyelinating peripheral nerve neuropathy [17], and one aimed to predict facial nerve function in acoustic neuroma patients [18] (Appendix A). A total of 5 studies were included in the final quantitative analysis. The diagram of the literature search is presented in Figure 1, whereas the details of data extraction from the included trials are listed in Table 1.

### 3.2. Study Characteristics 

The included studies involved a collective of approximately 840 participants. Of these participants, 1398 wrists underwent related analyses, and 718 writs were diagnosed with CTS. Diagnostic confirmation relied on electrophysiological assessments conducted in accordance with the guidelines set forth by the American Association of Neuromuscular and Electrodiagnostic Medicine [24]. The research flow of ultrasound radiomics in most included studies is presented in Figure 2. 

Furthermore, three [19,20,23] of these studies provided specific protocols for radiologists to evaluate the quality of median nerves, e.g., assessing nerve honeycomb pattern and echogenicity. In the context of these studies, radiomics features were extracted using MATLAB software (version R2019a, MathWorks Inc., Natick, MA, USA), Python, and Novo Ultrasound Kit (GE Institute of Precision Medicine, IPM) software, v. 3.12.0. The number of radiomic features analyzed in the included studies spanned from 10 to 369. When it comes to predictive models for CTS, all five studies employed machine learning techniques, with one study incorporating an additional deep learning approach. The machine learning for selection of hyperparameters encompassed various algorithms, such as support vector machines, regression models, random forests, K-nearest neighbors (KNN), and Bayesian methods. One of the included studies [23] utilized a deep learning model consisting of SqueezeNet.

It is worth noting that three of the included studies were conducted by the same research group. Their initial study [21] aimed to compare the impact of changes as regards the setting of region of interest setting (i.e., inclusion vs. exclusion of the epineurium) on the diagnostic performance of ultrasound radiomics. In their subsequent study [22], they explored the potential application of ultrasound radiomics in diagnosing mild CTS. Finally, their third study [20] focused on emphasizing the importance of utilizing specific characteristics of ultrasonic images for diagnosing CTS, without the need to measure their cross-sectional area. 

### 3.3. Quality Assessment

Table 2 shows the evaluation of methodological quality. Concerning the seven criteria in the QUADAS-2 tool, two studies [21,24] did not meet the criteria for assessing the risk of bias related to flow and timing. Only three studies [20,22,23] clearly defined a specific time interval between the electrodiagnostic test and the ultrasound examination. In three studies conducted by Lyu et al. [21,23,24], there was a lack of specification regarding exclusion criteria, which raised concerns about the applicability of their patient selection process.

### 3.4. Diagnostic Performance 

In the context of the radiologist evaluation, the collective statistics for sensitivity, specificity, and diagnostic odds ratio were as follows: 0.78 (95% CI, 0.71 to 0.83), 0.72 (95% CI, 0.59 to 0.81), and 9 (95% CI, 5 to 15), respectively (Figure 3A). The AUC obtained from the sROC curve analysis was 0.82 (95% CI, 0.78–0.85) (Figure 4A). Additionally, the Deeks’ funnel plot asymmetry test produced non-significant results, indicating the absence of substantial evidence for publication bias (*p* = 0.48) (Figure 5A).

Regarding the ultrasound radiomics training mode, the combined sensitivity, specificity, and diagnostic odds ratio were as follows: 0.88 (95% CI, 0.85 to 0.91), 0.88 (95% CI, 0.84 to 0.92), and 58 (95% CI, 38 to 87), respectively (Figure 3B). The AUC from the summary receiver sROC curve analysis was calculated to be 0.94 (95% CI, 0.91–0.96) (Figure 4B). Furthermore, the Deeks’ funnel plot asymmetry test generated non-significant results, suggesting the absence of substantial evidence of publication bias (*p* = 0.90) (Figure 5C).

Concerning the ultrasound radiomics testing mode, the aggregated sensitivity, specificity, and diagnostic odds ratio were as follows: 0.85 (95% CI, 0.78 to 0.89), 0.80 (95% CI, 0.73 to 0.85), and 22 (95% CI, 12 to 41), respectively (Figure 3C). The AUC derived from the summary receiver sROC curve analysis was determined to be 0.89 (95% CI, 0.85–0.91) (Figure 4C). Additionally, the Deeks’ funnel plot asymmetry test yielded non-significant results, indicating no substantial evidence of publication bias (*p* = 0.79) (Figure 5C).

## 4. Discussion

Our research stood as a pioneering effort in the field of neuromuscular ultrasound, as it represented one of the initial meta-analyses aimed at evaluating the effectiveness of ultrasound radiomics in diagnosing CTS. This work not only added an original dimension to the field but also addressed a critical gap in the utilization of ultrasound for CTS diagnosis. Significantly, our findings indicated that ultrasound radiomics outperformed assessments conducted by radiologists, underscoring its potential as a valuable diagnostic tool. Moreover, our research highlighted the remarkable consistency in diagnostic performance between the training and testing sets of ultrasound radiomics, reinforcing the method’s robustness and suitability for clinical applications in the realm of CTS diagnosis.

As addressed by a previous umbrella review [25], ultrasound has found place in the diagnosis of CTS. The pertinent ultrasonographic characteristics embrace nerve enlargement proximal to the compression site, flattening of the median nerve at the compression site, an increased hypoechoic portion due to elevated fluid content within the nerve, loss of fascicular pattern due to disruption of the internal nerve architecture, and increased vascularity as a sign of inflammation [26,27]. Among all these signs, the best and clinically useful indicator is the nerve cross-sectional area (CSA) measurement, using a cut-off value of 9–10.5 mm^2^ [25]. However, in patients with mild or chronic CTS, enlargement of the nerve cross-section may be less significant. In those cases, physicians heavily rely on the nerve’s shape or echotexture for the diagnosis. However, echotexture is subjective and difficult to standardize. Therefore, radiomics [1], which allows the extraction of various optic features from a given image, has been developed for the evaluation of nerve quality as well as the determination of entrapment neuropathy. 

A plethora of multidimensional radiometric features can be extracted from the presented images, far surpassing the capabilities of human visual perception. For instance, utilizing a common MATLAB module [28], we can harness the histogram to glean insights into the distribution of gray-level intensities. Additionally, the autoregressive model deciphers image texture by analyzing relationships between neighboring pixels, while the gradient feature captures variations in pixel intensities. The histogram of oriented gradients provides data about gradient occurrences in specific orientations, and the Gabor feature extracts frequency details from designated image directions. Wavelet analysis measures energy across sub-band images within specific high- and low-frequency channels, while the gray-level run-length matrix (GRLM) sheds light on pixel courses with identical intensity in specific directions. The gray-level co-occurrence matrix (GLCM) visually stands for pixel pair occurrences with specific intensities and directions. Lastly, local binary patterns (LBP) assess pixel intensity inequalities within two distinct neighborhood sizes. These quantitative features not only uncover intricate details beyond human perception but also facilitate standardization and comparison. This quantitative advantage likely underpins why ultrasound radiomics outperformed radiologist evaluation, which rely solely on visual assessment, in our meta-analysis.

During our review, we observed that the choice of the region of interest significantly impacts diagnostic performance. Lyu et al. [21] conducted a comparative analysis between the two methods, i.e., including and excluding the epineurium, where the latter demonstrated superior diagnostic accuracy. There are several factors that can explain these findings. First, the boundary between the epineurium and the interior nerve bundles was found to be clearer than the demarcation between the epineurium and exterior connective tissues. This clarity facilitates the standardization of nerve border delineation, leading to improved reliability across different investigators. Second, the primary sonographic pathologies of median nerves in CTS typically involve changes in the nerve fascicles [29]. Thickened epineurium is predominantly observed in cases of chronic CTS [8]. Consequently, including the epineurium in radiomic analysis provides unnecessary information during comparisons, which, in turn, adversely affects the diagnostic accuracy.

In the context of hyperparameter selection, our meta-analysis has identified various models, but the Support Vector Machine (SVM) emerged as the model demonstrating the most favorable diagnostic performance [30]. An SVM is a supervised machine learning model employed for both classification and regression tasks. It has a well-deserved reputation for its versatility and robustness across diverse applications, encompassing tasks such as image and text classification. The primary objective of an SVM is to discern an optimal hyperplane capable of effectively separating data points belonging to distinct classes while maximizing the margin between them. However, it is important to acknowledge that SVM often requires meticulous parameter tuning and can pose computational challenges, particularly when dealing with extensive datasets.

Another concern lies in the selection of multiple image features in radiomics, which can potentially contribute to model overfitting. Model overfitting is a common challenge in machine learning and statistical modeling, characterized by the model learning the training data too intricately. This leads to the model capturing not only underlying patterns and relationships but also noise and random fluctuations present in the data. Notably, the included studies in our analysis have implemented crucial strategies to mitigate overfitting. In this context, it is important to reiterate that the training mode is primarily focused on model development using a specific dataset, while the testing mode is geared towards assessing the model’s performance on new, unseen data, thereby evaluating its reliability and generalizability in clinical practice. For instance, they have employed techniques like the Minimum Redundancy Maximum Relevance method [31], which aims to maximize the correlation between features and CTS while minimizing the correlation between features and asymptomatic participants. These measures contribute to a more robust model. Furthermore, our meta-analysis has demonstrated that the diagnostic performance on the testing data is comparable to that on the training data, affirming the effectiveness of these efforts in preventing overfitting.

We need to acknowledge several limitations of our study. First, our inclusion criteria led to the incorporation of only five studies, with three of them being conducted by the same research group. Given the possibility of overlapping patient populations within these studies, the limited diversity in our dataset may have implications for the generalizability of our results. Second, manual selection of the region of interest was required in most of the included studies. This manual selection process raises concerns about standardization across different investigators. In the future, these limitations could potentially be addressed by implementing automatic segmentation methods using artificial intelligence [32]. Third, none of the included studies compared the diagnostic performance of ultrasound radiomics with the simpler measurement of nerve CSA. Indeed, CSA measurement has been established as a valuable ultrasound indicator for CTS. Therefore, future research should focus on conducting comparative studies between ultrasound radiomics and CSA measurements to better understand their relative effectiveness. Fourth, the participants’ age, sex, body mass index, and CTS severity could potentially influence the presentation of ultrasound images of the median nerve and impact diagnostic accuracy. However, it is essential to note that these details could not be sourced from the reviewed articles, and we acknowledge this as a limitation of our meta-analysis. Fifth, the available statistical test to detect publication bias in diagnostic meta-analysis, conducted through the ‘midas’ command using STATA, is the Deeks’ funnel plot asymmetry test. This test may have limitations due to its low power and susceptibility to false positives. It is essential to note that we did not utilize trim-and-fill analysis or Egger’s test, as these statistical methods were not compatible with the statistical approach employed in our study. Sixth, our evaluation of study quality unveiled a notable concern. Specifically, the examination of three studies conducted by Lyu et al. [21,23,24], indicated a lack of specification regarding their exclusion criteria. This omission raised significant questions about the transparency and applicability of their patient selection process. Moreover, there is a possibility that these three studies relied on the same dataset for their analyses. In light of these factors, it is prudent to exercise caution when interpreting the results obtained from our meta-analysis.

## 5. Conclusions

In contrast to assessments by radiologists, ultrasound radiomics displayed superior diagnostic performance in detecting CTS. Further, there seems to be little fluctuation in the diagnostic accuracy between the training and testing sets of ultrasound radiomics. Looking forward, there is exciting potential for improving the workflow of radiomics by incorporating automated segmentation techniques driven by artificial intelligence. In the future, it is essential to prioritize conducting comparative studies between ultrasound radiomics and CSA measurements to gain more profound understanding of their respective performance.

## Figures and Tables

**Figure 1 diagnostics-13-03280-f001:**
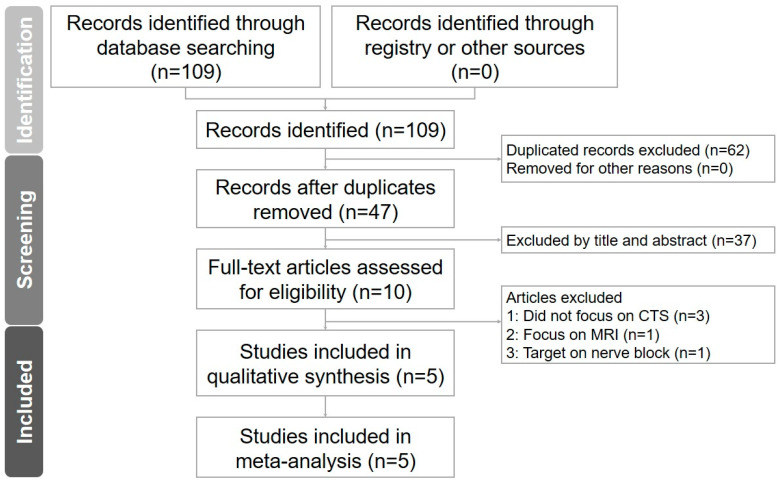
Flow diagram of the literature search.

**Figure 2 diagnostics-13-03280-f002:**
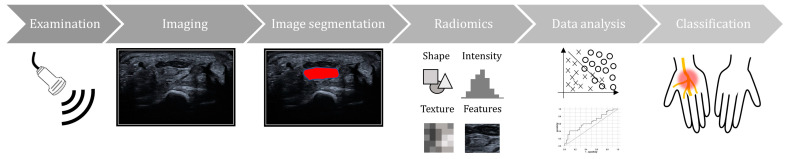
The research flow of ultrasound radiomic studies specific for carpal tunnel syndrome (CTS) diagnosis.

**Figure 3 diagnostics-13-03280-f003:**
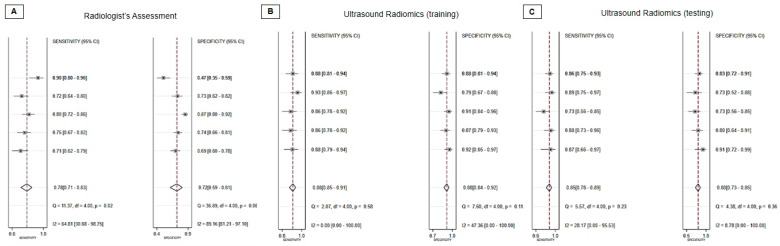
Forest plot of the summarized sensitivity and specificity of (**A**) radiologist assessment, (**B**) ultrasound radiomics training mode, and (**C**) ultrasound radiomics testing mode for the diagnosis of carpal tunnel syndrome.

**Figure 4 diagnostics-13-03280-f004:**
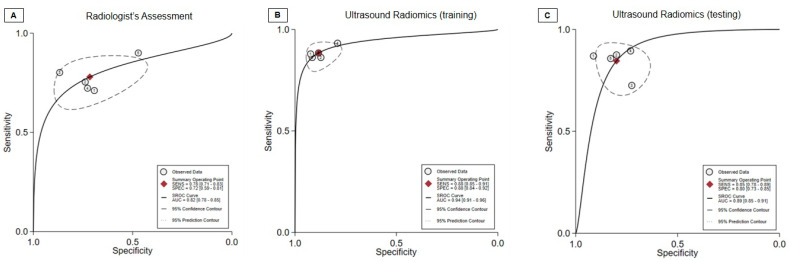
The summary receiver operating curve (sROC) curve of (**A**) radiologist assessment, (**B**) ultrasound radiomics training mode, and (**C**) ultrasound radiomics testing mode for diagnosing carpal tunnel syndrome. AUC, area under the curve; SENS, sensitivity; SPEC, specificity; SROC, summary receiver operating curve.

**Figure 5 diagnostics-13-03280-f005:**
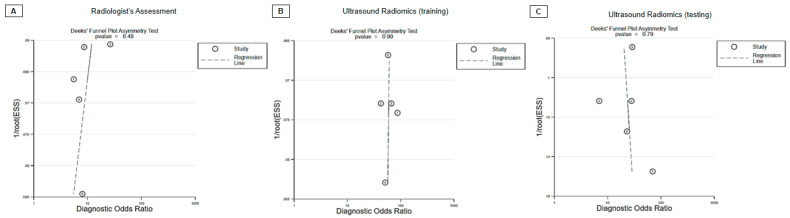
Deek’s Funnel Plot Asymmetry test of (**A**) radiologist assessment, (**B**) ultrasound radiomics training mode, and (**C**) ultrasound radiomics testing mode for diagnosing carpal tunnel syndrome.

**Table 1 diagnostics-13-03280-t001:** Characteristics of the included studies.

Author, Year	Country	Study Type	Mean Age of Participants (CTS/non-CTS)	% of Female Participants (CTS/non-CTS)	Ultrasound Machine/Transducer	Software	Hyperparameter	Model for Hyperparameter Selections
Faeghi et al., 2021 [19]	Iran	Prospective	51.93/50.28	85/84	AixPlorer Ultimate/5–18 MHz linear-array probe	MATLAB (version R2019a, MathWorks Inc.)	Histogram, AR model gradient, HOG, Gabor, wavelet, GLRLM, GLCM, LBP	SVM model
Lyu et al., 2023 a [20]	China	Retrospective	55.08/54.13	80/80	NA/7–15 MHz linear probe	ITK-SNAP 3.8	Maximum diameter, sphericity, entropy, median, joint entropy, contrast, correlation, short run low gray level emphasis, gray level non-uniformity, run entropy	Regression model
Lyu et al., 2023 b [21]	China	Retrospective	53.41/54.23	84/86	GE Vivid E9/15 MHz linear probe	Novo Ultrasound Kit (NUK)	Elongation, major axis length, maximum diameter, mesh surface, minor axis length, perimeter, perimeter surface ratio, pixel surface, sphericity, energy, total energy, entropy, 10 percentile, 90 percentile, maximum, minimum, mean, median, robust mean absolute deviation, root mean squared, skewness, kurtosis, uniformity, variance	Forest model
Lyu et al., 2023 c [22]	China	Retrospective	58.70/55.78	77/73	NA/15 MHz linear probe	ITK-SNAP 3.8	Sphericity, cluster prominence, short run low gray level emphasis, run entropy, long run high gray level emphasis, large area high gray level emphasis, gray level non uniformity, median, small area low gray level emphasis, size zone non uniformity, low gray level zone emphasis, size zone non uniformity normalized, kurtosis, major axis length	Regression model
Mohammadi et al., 2023 [23]	Iran and Colombia	Prospective	56.35/53.62	84/84	First center: AixPlorer Ultimate/12–18 MHz linear probe Second center: Min- dray MX7/15 MHz linear probe	Not mentioned	1000 deep radiomic features extracted by deep learning (SqueezeNet)	SVM model; SGD model; KNN model; GradBoost model; RForest model NBayes model; LogReg model; AdaBoost model DTree model

AR: autoregressive, HOG: histogram of oriented gradients, GLRLM: gray-level run-length matrix, GLCM: gray-level co-occurrence matrix, LBP: local binary patterns. SVM: support vector machine, NA: not available, SGD: stochastic gradient descent, KNN: k-nearest neighbors, GradBoost: gradient boosting, RForest: random forest, NBayes: naive Bayes, LogReg: logistic regression, AdaBoost: adaptive boosting, DTree: decision tree. Lyu et al., 2023, a: Application of ultrasound images-based radiomics in carpal tunnel syndrome: without measuring the median nerve cross-sectional area Lyu et al., 2023, b: ultrasound-based radiomics in the diagnosis of carpal tunnel syndrome: The influence of regions of interest delineation method on mode Lyu et al., 2023, c: The application of ultrasound image-based radiomics in the diagnosis of mild carpal tunnel syndrome.

**Table 2 diagnostics-13-03280-t002:** Methodological quality of the included studies assessed by QUADAS-2.

	Risk of Bias				Applicability Concerns		
Author, Year	Patient Selection	Index Text	Reference Standard	Flow and Timing	Patient Selection	Index Text	Reference Standard
Faeghi et al., 2021 [19]	Low	Low	Low	Low	Low	Low	Low
Lyu et al., 2023 a [20]	Low	Low	Low	High	High	Low	Low
Lyu et al., 2023 b [21]	Low	Low	Low	Low	High	Low	Low
Lyu et al., 2023 c [22]	Low	Low	Low	High	High	Low	Low
Mohammadi et al., 2023 [23]	Low	Low	Low	Low	Low	Low	Low

QUADAS, Quality Assessment of Diagnostic Accuracy Studies. Lyu et al., 2023, a: application of ultrasound images-based radiomics in carpal tunnel syndrome: without measuring the median nerve cross-sectional area. Lyu et al., 2023, b: ultrasound-based radiomics in the diagnosis of carpal tunnel syndrome: The influence of regions of interest delineation method on mode. Lyu et al., 2023 c: The application of ultrasound image-based radiomics in the diagnosis of mild carpal tunnel syndrome.

## Data Availability

The datasets used and/or analyzed during the current study are available from the corresponding author on reasonable request.

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
