# Peer review of "The Potential of Ultrasound Radiomics in Carpal Tunnel Syndrome Diagnosis: A Systematic Review and Meta-Analysis"

_diagnostics, 2023, doi:10.3390/diagnostics13203280_

Round 1

Reviewer 1 Report

Comments:

The paper is very interesting and well written.  I only have two comments:

1.      Please minimize the pronoun “we” especially in the statistical analysis section

2.     What is the difference between training mode and testing mode for radiomics.

Additional Comments:

1.     Do you consider the topic original or relevant in the field? Does it

address a specific gap in the field?

2.     Methods

a.     What was performed in order that the quality assessment by the two reviewers will be consistent?

b.     What was performed in order that the data extracted from the journals will be consistent  between the two reviewers?

c.      Why was the outcome of quality assessment did not influence the inclusion or exclusion of the article?

d.     What is the difference between training mode and testing mode for radiomics

3.      Results

a.     Three papers were  by Lyu et al.   How can the authors ensure that the subjects in the three papers were not the same?

b.     Please include any additional comments on the tables and figure 

4.     What does it add to the subject area compared with other published

material?

5.     Please minimize the pronoun “we” especially in the statistical analysis section

Author Response

Reviewer 1

Comments: 

The paper is very interesting and well written.  I only have two comments: 

  1. Please minimize the pronoun “we” especially in the statistical analysis section

Response:

    We are grateful for the reviewer's thoughtful feedback. In response to the reviewer's suggestion, we have reduced the usage of the pronoun "we" in the revised manuscript.

  1. What is the difference between training mode and testing mode for radiomics.

Response:

 We extend our gratitude to the reviewer for their insightful feedback. We have taken into account their suggestion and incorporated an explanation of the distinction between the training mode and testing mode for radiomics into the revised manuscript. In this context, we have included the following statement: "In this context, it's important to reiterate that the training mode is primarily focused on model development using a specific dataset, while the testing mode is geared towards assessing the model's performance on new, unseen data, thereby evaluating its reliability and generalizability in clinical practice"(line 326-329).

Additional Comments:

1. Do you consider the topic original or relevant in the field? Does it address a specific gap in the field?

Response:

  We are grateful for the reviewer's thoughtful comments. In response to the concerns raised, we have revised the first paragraph of the discussion to emphasize the originality and relevance of our research in the field of neuromuscular ultrasound. It now reads: “Our research stood as a pioneering effort in the field of neuromuscular ultrasound, as it represented one of the initial meta-analyses aimed at evaluating the effectiveness of ultrasound radiomics in diagnosing CTS. This work not only added an original dimension to the field but also addressed a critical gap in the utilization of ultrasound for CTS diagnosis. Significantly, our findings indicated that ultrasound radiomics outperformed assessments conducted by radiologists, underscoring its potential as a valuable diagnostic tool. Moreover, our research highlighted the remarkable consistency in diagnostic performance between the training and testing sets of ultrasound radiomics, reinforcing the method's robustness and suitability for clinical applications in the realm of CTS diagnosis”(line 259-267).

2. Methods

  • What was performed in order that the quality assessment by the two reviewers will be consistent?

Response:  The quality assessment and data extraction were carried out independently by two authors. Initially, there was no strict requirement for consistency in the initial stage of quality assessment. However, in the event of any inconsistencies concerning specific items, resolution was achieved through discussion between the two authors. Furthermore, such discrepancies were reported to the corresponding author for a final decision. The aforementioned statement has been added in the session describing “Quality Assessment” as “Discrepancies in the quality assessment were resolved through a consensus-based approach, mirroring the procedure followed for data extraction”(line 126-128).

  • What was performed in order that the data extracted from the journals will be consistent between the two reviewers?

Response:

The quality assessment and data extraction were carried out independently by two authors. Initially, there was no strict requirement for consistency in the initial stage of quality assessment. However, in the event of any inconsistencies concerning specific items, resolution was achieved through discussion between the two authors. Furthermore, such discrepancies were reported to the corresponding author for a final decision. The aforementioned statement has been added in the session describing “Inclusion and Exclusion Criteria” as “To ensure the rigor of our review process, two independent authors, W.-T.W. and T.-Y.L., initially conducted an evaluation of the titles and abstracts of the papers. Eligible articles underwent a comprehensive examination in their entirety to confirm their alignment with all the stipulated inclusion and exclusion criteria. The data extraction was performed independently by two authors. In the initial stage of data extraction, strict consistency was not mandated. However, in cases where inconsistencies arose regarding specific items, these were resolved through discussion between the two authors. Additionally, any unresolved discrepancies were reported to the corresponding author for a final decision”(line 109-120).

  • Why was the outcome of quality assessment did not influence the inclusion or exclusion of the article?

Response:

    We are grateful for the reviewer's feedback. It is essential to emphasize that the inclusion and exclusion of a study should strictly adhere to the pre-established criteria, which ideally should be publicly available on an open registry prior to commencing the literature review. Deviating from this approach by incorporating quality assessment results into the decision-making process for inclusion or exclusion could introduce bias into the outcome of a meta-analysis. We hope this clarification addresses the reviewer's concern.

  • What is the difference between training mode and testing mode for radiomics

Response:

 We extend our gratitude to the reviewer for the insightful feedback. We have taken the suggestion into account and incorporated an explanation of the distinction between the training mode and testing mode for radiomics into the revised manuscript. In this context, we have included the following statement: "In this context, it's important to reiterate that the training mode is primarily focused on model development using a specific dataset, while the testing mode is geared towards assessing the model's performance on new, unseen data, thereby evaluating its reliability and generalizability in clinical practice"(lie 326-329).

3. Results

  • Three papers were by Lyu et al.  How can the authors ensure that the subjects in the three papers were not the same?

Response: We value the reviewer's feedback. It is important to note that we cannot ascertain whether there was any overlap in patient populations among the three articles authored by Lyu et al. As a result, our discussion acknowledges this potential concern: "First, our inclusion criteria led to the incorporation of only five studies, with three of them being conducted by the same research group. Given the possibility of overlapping patient populations within these studies, the limited diversity in our dataset may have implications for the generalizability of our results"(line 336-340).

  • Please include any additional comments on the tables and figure.

Response: All the necessary comments have been added in the revised tables and figures.

4. What does it add to the subject area compared with other published material?

Response: We have emphasized the uniqueness of our research in the context of the subject area compared to previously published material: “Our research stood as a pioneering effort in the field of neuromuscular ultrasound, as it represented one of the initial meta-analyses aimed at evaluating the effectiveness of ultrasound radiomics in diagnosing CTS. This work not only added an original dimension to the field but also addressed a critical gap in the utilization of ultrasound for CTS diagnosis. Significantly, our findings indicated that ultrasound radiomics outperformed assessments conducted by radiologists, underscoring its potential as a valuable diagnostic tool. Moreover, our research highlighted the remarkable consistency in diagnostic performance between the training and testing sets of ultrasound radiomics, reinforcing the method's robustness and suitability for clinical applications in the realm of CTS diagnosis”(line 259-267).

5. Please minimize the pronoun “we” especially in the statistical analysis section

Response: We are grateful for the reviewer's thoughtful feedback. In response to the reviewer's suggestion, we have reduced the usage of the pronoun "we" in the revised manuscript.

Reviewer 2 Report

Dear Authors, Thank you for the opportunity to read the results of the submitted articlew. I consider topic important and interesting. The article is well prepared methodologically, the statistical analysis complements the research results. The only drawback is, as the authors themselves emphasize, the small number of available manuscripts, which does not reduce the value of the article.

best regards

Author Response

Reviewer 2

Comment:

Dear Authors, thank you for the opportunity to read the results of the submitted article. I consider topic important and interesting. The article is well prepared methodologically, the statistical analysis complements the research results. The only drawback is, as the authors themselves emphasize, the small number of available manuscripts, which does not reduce the value of the article.
best regards

Response:  

We would like to express our gratitude for the reviewer's valuable input, particularly for acknowledging the effort we invested in this manuscript. As the reviewer rightly noted, the primary limitation lies in the relatively small number of studies included. We remain committed to addressing this concern. Should additional pertinent articles become available in the future, we will explore the possibility of performing an updated meta-analysis to further validate the effectiveness of ultrasound radiomics in diagnosing CTS.

Reviewer 3 Report

The novelty of this study is that it is the first meta-analysis to compare the diagnostic accuracy of ultrasound radiomics with other methods for carpal tunnel syndrome diagnosis. Ultrasound radiomics is a novel technique that extracts and analyzes quantitative features from ultrasound images that can reveal hidden patterns and associations in medical imaging data. The study claims that ultrasound radiomics has superior diagnostic performance than radiologist evaluation and shows minimal variability between training and testing sets. There are some comments that should be considered by the authors:

1. Introduction

·       While the authors mention their objective towards the end of the introduction, it could be beneficial to state the research question more explicitly. For instance, “This meta-analysis aims to investigate: ‘What is the diagnostic accuracy of ultrasound radiomics in detecting Carpal Tunnel Syndrome compared to other diagnostic methods?

·       The introduction could benefit from a clearer explanation of why a meta-analysis is needed in this context. If there are conflicting results from previous studies or if this is a novel area of study, that should be stated.

·       The authors have provided some background on Carpal Tunnel Syndrome (CTS) and ultrasound radiomics. However, they could consider providing more context about why ultrasound radiomics might be particularly useful for CTS diagnosis and how it compares to other methods.

·       The authors might want to elaborate on why this meta-analysis is important. Who will it impact? How will it contribute to the field?

2. Methods

·       The authors should report how they extracted and verified the data from the included studies, such as the variables, outcomes, and covariates of interest, and how they handled missing or incomplete data.

·       The authors should explain how they defined and measured the intervention (ultrasound radiomics) and the comparison (evaluation by radiologists) in the inclusion and exclusion criteria subsection. They should also specify if they included studies with different types of ultrasound radiomics or radiologist evaluations, and how they dealt with any heterogeneity or variability among them.

·       The authors should describe how they addressed the sensitivity of their results

3. Results

·       The authors should provide more details about the characteristics of the participants, such as their age, sex, BMI, and CTS severity. This would help to assess the generalizability and heterogeneity of the studies.

·       The authors should provide more details about the characteristics of the participants, such as their age, sex, BMI, and CTS severity. This would help to assess the generalizability and heterogeneity of the studies.

·       The authors should provide more details about the characteristics of the participants, such as their age, sex, BMI, and CTS severity. This would help to assess the generalizability and heterogeneity of the studies.

·       The authors should discuss the limitations of using Deeks’ funnel plot asymmetry test to detect publication bias, such as its low power and susceptibility to false positives. They should also consider using other methods to explore publication bias, such as trim-and-fill analysis or Egger’s test.

·       The authors should provide a summary of the main findings and implications of the quality assessment in the text, not just in Table 2. They should also explain how they dealt with the risk of bias and applicability issues in their analysis and interpretation of the results.

·       The authors should provide more details about the radiologist evaluation, ultrasound radiomics training mode, and ultrasound radiomics testing mode. They should explain what these terms mean, how they were performed, and what were the differences and similarities among them.

The English language quality of this study is generally good and some minor modifications in a final review should be sufficient

Author Response

Reviewer 3

The novelty of this study is that it is the first meta-analysis to compare the diagnostic accuracy of ultrasound radiomics with other methods for carpal tunnel syndrome diagnosis. Ultrasound radiomics is a novel technique that extracts and analyzes quantitative features from ultrasound images that can reveal hidden patterns and associations in medical imaging data. The study claims that ultrasound radiomics has superior diagnostic performance than radiologist evaluation and shows minimal variability between training and testing sets. There are some comments that should be considered by the authors:

Response: We are grateful for the reviewer's feedback, and we have diligently incorporated their suggestions into the revised article.

  1. Introduction
  • While the authors mention their objective towards the end of the introduction, it could be beneficial to state the research question more explicitly. For instance, “This meta-analysis aims to investigate: ‘What is the diagnostic accuracy of ultrasound radiomics in detecting Carpal Tunnel Syndrome compared to other diagnostic methods?

Response: We are grateful for the reviewer's feedback. The objective of this meta-analysis has been revised as “Due to the lack of previous meta-analyses on this subject, our objective was to investigate the diagnostic accuracy of ultrasound radiomics in detecting CTS and to compare it with other ultrasound diagnostic approaches”(line 71-73).

  • The introduction could benefit from a clearer explanation of why a meta-analysis is needed in this context. If there are conflicting results from previous studies or if this is a novel area of study, that should be stated.

Response: We appreciate the reviewer's feedback. It's important to clarify that the primary motivation for this meta-analysis was not to reconcile conflicting evidence from various studies, as such evidence did not exist at the time. Instead, our main impetus was the absence of any prior meta-analysis on this topic. Given that CTS represents the most prevalent nerve entrapment syndrome, we recognized the significance of conducting this meta-analysis. This rationale is succinctly articulated in our stated objective: "“Due to the lack of previous meta-analyses on this subject, our objective was to investigate the diagnostic accuracy of ultrasound radiomics in detecting CTS and to compare it with other ultrasound diagnostic approaches” (line 71-73).

  • The authors have provided some background on Carpal Tunnel Syndrome (CTS) and ultrasound radiomics. However, they could consider providing more context about why ultrasound radiomics might be particularly useful for CTS diagnosis and how it compares to other methods.

Response: We are thankful for the reviewer's feedback. We have incorporated the following statement into the introduction: "The potential benefit of ultrasound radiomics in diagnosing CTS lies in its capacity to extract and analyze intricate image features, thereby providing a more detailed and objective assessment of the condition when compared to the sole estimation of the nerve's cross-section or shape"(line 68-71).

  • The authors might want to elaborate on why this meta-analysis is important. Who will it impact? How will it contribute to the field?

Response: The meta-analysis aims to validate whether radiomics can enhance the diagnostic efficacy of standard ultrasound imaging for the evaluation of CTS, making a substantial contribution to the field of neuromuscular pain medicine. This statement has been included in the revised manuscript. The added part reads: “The meta-analysis would aid in validating whether radiomics could improve the diagnostic efficacy of standard ultrasound imaging for assessing CTS, thereby making a significant contribution to the field of neuromuscular pain medicine” (line 74-76).

  1. Methods
  • The authors should report how they extracted and verified the data from the included studies, such as the variables, outcomes, and covariates of interest, and how they handled missing or incomplete data.

Response: We appreciate the kind comments from the reviewer. We have added a paragraph to explain how we extracted and verified the data as: “To ensure the rigor of our review process, two independent authors, W.-T.W. and T.-Y.L., initially conducted an evaluation of the titles and abstracts of the papers. Eligible articles underwent a comprehensive examination in their entirety to confirm their alignment with all the stipulated inclusion and exclusion criteria. The data extraction was per-formed independently by two authors. In the initial stage of data extraction, strict consistency was not mandated. However, in cases where inconsistencies arose regarding specific items, these were resolved through discussion between the two authors. Additionally, any unresolved discrepancies were reported to the corresponding author for a final decision. In the event of missing or incomplete data, our approach would involve reaching out to the corresponding author of the referenced paper. Should we not receive a response from the corresponding authors, we would document this situation in supplementary tables”(line 109-120).

  • The authors should explain how they defined and measured the intervention (ultrasound radiomics) and the comparison (evaluation by radiologists) in the inclusion and exclusion criteria subsection. They should also specify if they included studies with different types of ultrasound radiomics or radiologist evaluations, and how they dealt with any heterogeneity or variability among them.

Response:

     We would like to express our gratitude for the kind comment provided by the reviewer. In response to their feedback, we have introduced a comprehensive definition of ultrasound radiomics, stating, "Ultrasound radiomics is a quantitative and data-driven approach that involves the extraction and analysis of a wide range of features from ultrasound images. This process yields valuable insights into tissue characteristics and potential disease markers" (line 97-100). Furthermore, the term "comparison" has been more explicitly defined in our work as "The assessment of ultrasound imaging for the diagnosis of CTS by radiologists"(line 95-96). It is important to note that our study did not focus on specific types of ultrasound radiomics or on the evaluation conducted by radiologists. We appreciate the reviewer's diligence in highlighting this point. To address the issue of heterogeneity, we have detailed our approach to assessing it in the statistical analysis section, stating, "Additionally, we assessed heterogeneity using I2 statistics and deemed significant heterogeneity present when I2 exceeded 50%"(line 151-153).

    The paragraph describing the PICO question has been revised as “The PICO (population, intervention, comparison, and outcome) question was struc-tured in the following manner: (1) Population: Individuals diagnosed with CTS; (2) Inter-vention: Application of ultrasound radiomics; (3) Comparison: Evaluation of ultrasound imaging for the diagnosis of CTS by radiologists and (4) Outcome: Assessment of diagnos-tic performance indicators, such as sensitivity and specificity. In the context, ultrasound radiomics is a quantitative and data-driven approach that involves extracting and ana-lyzing a wide range of features from ultrasound images to provide valuable insights into tissue characteristics and potential disease markers. Furthermore, in our mete-analysis, we did not specify any specific types of ultrasound radiomics or radiologist’s evaluation”(line 93-102).

  • The authors should describe how they addressed the sensitivity of their results

Response:

  We appreciate the kind comment from the reviewer. The elaboration of sensitivity and specificity have been added in the revised manuscript as “Sensitivity assesses the diagnostic test's capacity to accurately detect individuals, such as those assessed using ultrasound radiomics or radiologist's evaluation, who have a specific condition or disease, like the presence of CTS. It quantifies the proportion of true positive results among all individuals who genuinely have the condition. In contrast, specificity gauges the test's ability to correctly identify individuals without the condition, indicating the proportion of true negative results among all individuals who are indeed free of the condition”(line 138-144).

  1. Results
  • The authors should provide more details about the characteristics of the participants, such as their age, sex, BMI, and CTS severity. This would help to assess the generalizability and heterogeneity of the studies.

Response: We extend our gratitude to the reviewer for their feedback. Regrettably, we were unable to obtain data regarding participants' age, sex, body mass index, and CTS severity from the included articles, as this information was not available. We have duly acknowledged this limitation in our revised manuscript, stating, "The participants' age, sex, body mass index, and CTS severity could potentially influence the presentation of ultrasound images of the median nerve and impact diagnostic accuracy. However, it is essential to note that these details could not be sourced from the reviewed articles, and we acknowledge this as a limitation of our meta-analysis"(line 348-352).

  • The authors should discuss the limitations of using Deeks’ funnel plot asymmetry test to detect publication bias, such as its low power and susceptibility to false positives. They should also consider using other methods to explore publication bias, such as trim-and-fill analysis or Egger’s test.

Response:

The statistical software employed for conducting meta-analysis is STATA. In the realm of diagnostic meta-analysis, the tool for detecting publication bias, as implemented through the 'midas' command, is Deeks’ funnel plot asymmetry test. It's worth noting that this test may have limitations due to its lower statistical power and susceptibility to false positives.

However, it's important to clarify that we did not employ the trim-and-fill analysis or Egger's test. The reason for this choice is that these two statistical tests were not supported by the specific statistical approach we utilized. This information has been incorporated into the limitations section of our study, where we state, "Fifth, the available statistical test to detect publication bias in diagnostic meta-analysis, conducted through the 'midas' command using STATA, is Deeks’ funnel plot asymmetry test. This test may have limitations due to its low power and susceptibility to false positives. It's essential to note that we did not utilize trim-and-fill analysis or Egger's test, as these statistical methods were not compatible with the statistical approach employed in our study"(lie 352-357).

  • The authors should provide a summary of the main findings and implications of the quality assessment in the text, not just in Table 2. They should also explain how they dealt with the risk of bias and applicability issues in their analysis and interpretation of the results.

Response:

    We appreciate the comment from the reviewer. Regarding the main findings, we have summarized in the first paragraph of the discussion as “Our research stood as a pioneering effort in the field of neuromuscular ultrasound, as it represented one of the initial meta-analyses aimed at evaluating the effectiveness of ultra-sound radiomics in diagnosing CTS. This work not only added an original dimension to the field but also addressed a critical gap in the utilization of ultrasound for CTS diagno-sis. Significantly, our findings indicated that ultrasound radiomics outperformed assess-ments conducted by radiologists, underscoring its potential as a valuable diagnostic tool. Moreover, our research highlighted the remarkable consistency in diagnostic performance between the training and testing sets of ultrasound radiomics, reinforcing the method's robustness and suitability for clinical applications in the realm of CTS diagnosis”(line 259-267).

   As for the results of our quality assessment, we refrained from conducting subgroup analysis due to the limited number of studies encompassed in this meta-analysis. However, we concurred with the reviewer regarding the critical importance of considering study quality in our interpretation. Consequently, we incorporated the following statement into the discussion section: “Sixth, our evaluation of study quality unveiled a notable concern. Specifically, the exam-ination of three studies conducted by Lyu et al. [21,23,24], indicated a lack of specification regarding their exclusion criteria. This omission raised significant questions about the transparency and applicability of their patient selection process. Moreover, there is a pos-sibility that these three studies relied on the same dataset for their analyses. In light of these factors, it is prudent to exercise caution when interpreting the results obtained from our meta-analysis”(line 357-364).

  • The authors should provide more details about the radiologist evaluation, ultrasound radiomics training mode, and ultrasound radiomics testing mode. They should explain what these terms mean, how they were performed, and what were the differences and similarities among them.

Response:

We value the reviewer's input. To address the concern regarding radiologist evaluation, we have made it explicit in the PICO question: Evaluation of ultrasound imaging for the diagnosis of CTS by radiologists”(line 95-96). Regarding the distinction between the training and testing modes, we have provided further clarification: “In this context, it's important to reiterate that the training mode is primarily focused on model development using a specific dataset, while the testing mode is geared towards assessing the model's performance on new, unseen data, thereby evaluating its reliability and generalizability in clinical practice”(line 326-329). .